# The Effect of AlI_3_ Nanoadditive on the Thermal Behavior of PMMA Subjected to Thermoanalytical Py-GC-MS Technique

**DOI:** 10.3390/ma14227036

**Published:** 2021-11-19

**Authors:** Muhammad Adnan, Taj Ur Rahman, Ali Bahadur, Muhammad Aurang Zeb, Wajiha Liaqat, Takashiro Akitsu, Shams H. Abdel-Hafez, Wael A. El-Sayed

**Affiliations:** 1Department of Chemistry, Mohi-ud-Din Islamic University, Nerian Sharif 12080, AJ&K, Pakistan; adnanjspsc@gmail.com (M.A.); gia_248@yahoo.com (W.L.); 2Department of Transdisciplinary Studies, Graduate School of Convergence Science and Technology, Seoul National University, Seoul 08826, Korea; 3Department of Chemistry, Faculty of Science Division II, Tokyo University of Science, Tokyo 162-8601, Japan; akitsu2@rs.tus.ac.jp; 4Department of Chemistry, College of Science, Taif University, P.O. Box 11099, Taif 21944, Saudi Arabia; s.abdelhafez@tu.edu.sa; 5Department of Chemistry, College of Science, Qassim University, Buraidah 51452, Saudi Arabia; w.shendy@qu.edu.sa

**Keywords:** PMMA, aluminum triiodide, thermoanalytical study, degradation

## Abstract

Thermal degradation of polymethylmethacrylate (PMMA) was studied by using inorganic salt of aluminum triiodide (AlI_3_). The composites of PMMA were prepared with AlI_3_ by changing the concentration of the AlI_3_ additive from 2% to 10% (*w*/*w*). The PMMA composites with AlI_3_ were characterized by TGA, DTG, SEM, FTIR, HBT, and Py-GC-MS techniques. The FTIR peaks of PMMA composite at 1316, 786, and 693 cm^−1^ justify the chemical association between PMMA and AlI_3_. TGA study shows that the stability of PMMA is enhanced by the addition of the AlI_3_ additive. SEM analysis represented that there is a relationship between polymer and additive when they are mixed at the molecular level. The horizontal burning test (HBT) also confirmed that the AlI_3_ additive produced the flame retarding properties in PMMA polymer. The burning rate of composite with 10% of AlI_3_ additive decreases five times as much as compared to pure PMMA polymer. Py-GC-MS analysis deduced that PMMA composite produced less toxic and environment-friendly substances (CO_2_) by the influence of AlI_3_ additive as compared to neat PMMA.

## 1. Introduction

Waste plastics are usually disposed of in landfill sites or utilized energetically, for instance, by incineration [1]. The thermal degradation pattern of the different polymeric systems has been studied extensively due to their importance in everyday life [2,3] The thermal study of polymers gives important information about stability, stiffness, toughness, and types of residues produced [4,5]. The pathway of the degradation of commercial polymers has been reported by many researchers [6].

PMMA is an amorphous polymer related to the acrylate family; it is transparent, tough, and thermoplastic, has excellent weather ability, is well compatible with human tissue, and has impact resistance [7]. PMMA is also used for the welfare of mankind in the form of architecture, construction, automotive, transportation, lighting, electronics, photoelectricity, medical and healthcare, furniture, solar and nanotechnology applications, sensor application, and optical applications [8,9]. The annual production of PMMA plastics in Korea amounts to 100,000 tons [10]. PMMA has undergone combustion reactions giving the various volatile and nonvolatile products which cause environmental pollution [11]. Various analysis methods are used to detect the degradation behavior of polymers [12]. It resulted that the polymer showed a remarkably different thermal pattern when heated in combination with a small amount of additive [13]. Both components of the system were shown a certain physical and chemical association. The thermal degradation products of neat polymers were different as compared to the polymer-additive system produced [14]. Various organic and inorganic fire retardants were used in the polymer industry. The thermal behavior of PMMA alone [15] and with different additives has been extensively addressed by many workers [16].

The investigation of the effect of metallic halides on PMMA characteristics and the thermal degradation behavior was carried out in this study. To our knowledge, only a few studies had comprehensively addressed the thermal degradation behavior of PMMA polymers. Previously, the thermal degradation of PMMA was investigated using various reactors such as vessels, autoclaves, rotary kilns, and fluidized beds [17]. In this study, we designed a pyrolytic assembly to study the thermal degradation of PMMA under an inert environment. The present work studies the thermal behavior of PMMA in the presence of inorganic metallic halide additives such as AlI_3_, which provides stabilization to the PMMA in variable temperature zones. Certain new products in a higher yield were also noticed, which were absent when the polymer was degraded alone. The obtained degradation products were characterized by employing FTIR, SEM, TGA, DTG, HBT, and Py-GC-MS techniques.

## 2. Materials and Methods

### 2.1. Reagents

All chemicals taken from standard source suppliers were of analytical grade. Aluminum triiodide (Aldrich Sigma, St. Louis, MO, USA; 99% purity), methyl methacrylate monomer (E. Merck, St. Louis, MO, USA), 2, 2′-azobisisobutyronitrile (AIBN, E. Merck), methanol, and acetone were used as received.

### 2.2. Synthesis of Poly (Methyl Methacrylate)

The PMMA was synthesized by a free radical polymerization process in 250 mL three-neck reaction flasks using a free radical initiator (AIBN). The MMA monomer (10 mL) was dissolved in acetone and added to the reaction flask. AIBN (0.1% (*w*/*w* dissolved in acetone was added into the MMA solution dropwise, and polymerization proceeded under a hot water bath for 3 h at 70 °C. The final PMMA solution was poured into toluene and then precipitated out polymer by using methanol [18,19]. The PMMA was separated by vacuum filtration to obtain a dry polymer.

### 2.3. Synthesis of PMMA Composite with AlI_3_

PMMA and aluminum triiodide composite with various ratios was obtained in the form of a thin film by using acetone as a common solvent (Table 1). The calculated amount of polymer and additive were mixed individually in a fixed volume of acetone. Both mixtures were mixed in a sealed Pyrex tube by continuously stirring at ambient temperature and then poured into a Petri dish [6]. The obtained films were dried at room temperature and stored in a desiccator for further characterization.

### 2.4. Preparation of Strip for HBT

The PMMA and their composite were put in the common solvent (acetone) overnight to dissolve completely. The solution was placed in the Teflon molds and dried for 48 h to form strips of a specific length (125 mm × 7 mm × 1 mm) [20]. The final dried strips were kept in a desiccator for further study.

### 2.5. Horizontal Burning Test (HBT)

HBT of PMMA composites with AlI_3_ was conducted according to ASTM D 4986. One end of the PMMA blend stirp was attached horizontally to a stand, and the other end of the strip was ignited with a flame. The time of flame move from 1st reference, marked as A (25 mm from the end) to another reference, marked as B (100 mm from the same end), was noted.

### 2.6. Pyrolytic Study of PMMA Composite

The composite M5 (90% PMMA, 10% AlI_3_) was subjected to pyrolytic study on heating at high temperature in pyrolytic glass assembly. The volatile degradation products were collected in acetone at −196 °C temperature that was maintained under a nitrogen atmosphere. The separation of degradation products occurred by directly injecting a handsome volume of samples into a GC-MS instrument.

### 2.7. Instrumentation

Thermoanalytical determinations were obtained by utilizing NETZSCH simultaneous STA 429 thermal analyzer. Then, the thermal instability of the polymer sample was observed from 25 to 600 °C temperature in a nitrogen environment at a heating rate of 5 °C/min. FT-IR spectrometer (Nicolet 6700) was used within a range of 400–4000 cm^−1^, resolution 2 cm^−1^, 32 scans per measurement to gain FTIR spectra of polymer, additive, and residue obtained after ignition of the composite at various temperature zones. The GC-MS system employed was 5973 inert MSD combined with Agilent 6890N type through Agilent Analytical tool and Agilent Technologies, Santa Clara, CA, USA. The sample was run by using a DB-5 MS column filled with a volume of one microliter. The Tuscan XM 5136 instrument was used for SEM analysis of PMMA composite film.

## 3. Results

### 3.1. Thermoanalytical Study

The TGA and DTG thermograms of pure polymer, additive, and composites (M1–M5) were recorded under N_2_ atmosphere from 25 to 700 °C (Figure 1 and Figure 2). The comparative study of TGA and DTG and the data for polymer (M), additive (A), and polymer additive composite (M1–M5) are shown in Table 2. The kinetic data of composites M1–M5 are given in Appendix A.

TGA data of pure additive (AlI_3_) consist of a two-step thermal degradation. The first-step decomposition starts around temperature at 60 °C and is completed at 160 °C with a 20% weight loss, whereas a total weight loss 5% to 6% is indicated due to the removal of water vapor. The second-step decomposition starts at 165 °C and is completed at 490 °C with a 72% weight loss. Both decomposition steps show the production of I, I_2_, and I. (iodide free radical). Two DTG peaks indicate a two-step degradation process. At the end of the decomposition process, residue (8%) was found.

The thermal degradation pattern of M occurs in two steps. The first-step weight loss starts at 250 °C and is completed at 320 °C. In the first step of degradation, 10% weight loss is observed. It indicates the loss of the MMA monomer repeating unit without the release of residue at the end. In this system, the degradation pattern is caused by the separation of the end chain as well as the main chain breakage. The second stage of degradation starts at 320 °C and ends at 450 °C. In this step of degradation, weight loss (90%) is found without residue left. In composite M2, two DTG peaks appear at 290 and 390 °C, which indicate the two-step degradation process starts at 140 °C and was completed at 250 °C with a 10% weight loss. The products released indicate a certain association develops among both components of the composite. It was found that polymer M contains T_0_ at 250 °C, whereas additive commences decaying at 60 °C when separately heated in nitrogen condition. Thus, this is the second indication of certain associations among components of the system. Therefore, intermediates show instability at the end of first stage degradation. M polymer shows two DTG peaks at 290 and 390 °C, which confirms the two-step degradation process, whereas the second-step decomposition is accomplished at 460 °C with a 90% weight loss.

The thermal degradation of composite M1 occurs in two steps. The first-step degradation starts at 140 °C and ends at 250 °C with a 10% weight loss. The products released indicate a certain association among the components of the composite. Thus, intermediates show instability at the end of first-stage degradation. Two DTG peaks at 200 and 370 °C confirm the two-step degradation process, whereas the second-step decomposition starts at 250 °C and ends at 460 °C with an 87% weight loss. In the end, 2% of the residue is observed. The second DTG peak at 390 °C indicates the final stage of degradation. The sudden weight loss for the second stage of degradation indicates the rupture of all types of bonds as the rising energy content cannot be resisted.

The degradation of composite M2 starts at 130 °C and ends at 240 °C with a weight loss of 12%. By the increasing amount of additive (AlI_3_), T_0_ is the same when compared with [3] composite M1, but weight loss (2%) is increased. The same nature of the association is identified for this composite, as was noted for composite M1. The second stage of degradation starts at 240 °C and ends at 460 °C with a weight loss of 82%. The temperature ranges from 250 to 350 °C, and a 10% weight loss is observed due to some hindrance offered by certain associations exposed in the start portion of decomposition among both components of the composite, whereas the remaining 72% weight loss occurs on heating at 370–460 °C. At the end of the decomposition process, residue (6%) was found.

The decomposition of composite M3 starts at 140 °C and ends at 250 °C with a 14% weight loss. There are certain short-lived species formed in that step of the degradation process. The second stage decomposition starts at 250 °C and ends at 470 °C with a weight loss of 78% being recorded [21]. Their decrease in weight loss indicates a powerful association developed among the components of the system at the initial portion of the degradation process. There are two DTG peaks observed for first-step degradation at 180 and 370 °C. At the end of the decomposition process, residue (8%) was found.

The thermal decomposition of composite M4 proceeds in two steps. The first step of degradation starts at 115 °C and ends at 260 °C with a weight loss of 16%. This stage of degradation occurs by decomposition of the additive because the additive starts to decompose early. The additive decomposition occurs at an early stage due to the presence of an association between additive and pendant groups of the polymer, and heat exchange occurs [22]. This part of degradation indicates the instability of polymer and composite. It also gives a clue about the certain physical and chemical association between components of the composite system. The destabilization of the polymer system occurs by the production of iodine atoms (I) and iodine free radicals (I.). The second-step degradation starts at 260 °C and is completed at 470 °C with a 72% weight loss occurring. In this step, a slow rate of degradation is observed due to the presence of certain interactions, which resists the fast weight loss of the composite matrix. Thus, around 450 °C, the energy content suppresses the interaction, and the composite system degrades easily. The pyrolysis process is completed at 470 °C with the release of several the degradation products, along with that product which is not produced when a component of composite degrades separately with 12% found. The residue was identified as aluminum metal connected with oxygen and carbon aluminum metal [23].

The thermal decomposition of composite M5 is completed in two stages. The first step starts at 110 °C and ends at 270 °C with an 18% weight loss observed. The thermoanalytical study gives two DTG peaks at 130 and 350 °C. Some intermediate species are produced and suddenly decompose; therefore, the energy of the system increases. The second-step decomposition starts at 270 °C and ends at 470 °C with a 67% weight loss occurring. The first weight loss 7% of the weight needs heating at 130 °C (from 370 to 470 °C), and the remaining portion instead of the remaining 66% weight loss occurs at 20 °C (470–490 °C). This indicates a powerful association developed among the components of the system at the initial portion of the degradation process. This gives a clue of the strength of the bonds present in the intermediate.

The thermoanalytical study revealed that the first-step degradation of composite (M1 to M5) starts at a relatively lower temperature than neat polymer (M). It was found that by increasing the percentage of additives in the composites from 2% to 10%, degradable products are released at the temperature range of 140–250 °C and 110–270 °C with a loss of weight of 10–18%, respectively, whereas the second-step degradation of composite (M1–M5) volatile compound weight loss decreases from 87% to 67% and residue weight increases from 3% to 15% produced due to the development of certain interaction among components of the composite. The respective data and steps of thermal decomposition behavior of additive, polymer, and their composite are given in Table 2.

### 3.2. FTIR Analysis

FTIR study of the pure additive gives identical stretching for AlI_3_. Due to the hygroscopic nature of the additive, a broad peak is identified in the range of 2750–3250 cm^−1^, which represents the presence of water. The other peaks are observed at 1753, 1550, 1150, 1389, 1050, 750, 650, and 500 cm^−1^ due to Al-I linkage. The peak around 1535–1635 cm^−1^ indicates the ester functional group linkages for M. The region of spectra 1630–1640 cm^−1^ gives no peak, which shows the transformation to a polymeric form of the compound and the absence of unsaturation. The saturated C-H stretching was identified around 3100 cm^−1^. The FTIR spectra of the composite system (M1–M5) were identified with a different concentration ratio of polymer (PMMA) to additive (AlI_3_). It is representative of peaks for PMMA to have altered positions. This may be due to the formation of the composite and certain physical associations being developed between components of the composite. For composites, the peaks for ester functional group bonds of PMMA appear at the lower frequency at 1616 cm^−1^ [24]. whereas Al-I stretching is either absent or appears at a higher frequency. The composite system peaks identified at 1316, 786, and 693 cm^−1^ are identified as the certain interaction between AlI_3_ and PMMA, as shown in Figure 3. Both PMMA and AlI_3_ were soluble in acetone and mixed at the molecular level.

### 3.3. Py-GC-MS Analysis

The composite M5 (90% PMMA, 10% AlI_3_) was subjected to pyrolytic study on heating at high temperature in pyrolytic glass assembly. In GC-MS, various degradation products were separated based on their retention time. GAS chromatography has represented the result in the form of various peaks. Different peaks of degradation products were plotted in abundance against time. The MS of each GC peak was measured on the calculation of m/e value. Therefore, these results were matched with the available data in the library system. The GC-MS of this composite represents five peaks. The identified products that have represented the relationship among the constituents of the system are shown in Figure 4.

Peak 1 was identified as an aldehydic compound. It was believed that AlI_3_ made it easy to separate –OCH_3_ group from the pendant unit of PMMA. Al and iodine appeared to be the essential part of the completely degraded unit of PMMA. Peak 2 was identified as the presence of an additive due to holding the aldehyde functional group. Iodine was attached with the backbone as the whole pendant group (COOCH_3_) was removed. Only two neighboring MMA units proceeded with these changes, i.e., the conversion of the ester to aldehyde and attachment of iodine. Peak 3 represents how the MMA unit remains attached to the compound. For the adjacent part, only unsaturation was notified with iodine connected to terminal carbon. CH_3_ and COOCH_3_ might degrade CH_4_ and CO_2_.

Peak 4 shows that all the iodine of AlI_3_ was found to be substituted by COOCH_3_ from the MMA unit. Peak 5 identified that two ether groups (OCH_3_) had been substituted by iodine and AlH_2_, whereas one methyl of remaining OCH_3_ through H. One of the CH_3_ of the backbone seems to have been substituted by iodine. All these changes were affected by the presence of the additive. The conversion of AlI_2_ to AlH_2_ suggests the ease with which hydrogens diffuse through the system and substitute the iodine of aluminum. The formulae and e/m of all identified compounds are shown in Table 3. The GC-MS study made it easy to understand the degradation process of the composite system. The association between additive and polymer imparts the instability to the system in certain parts. The mixing of AlI_3_ in the system is affected by the production of the products identified during the degradation at various temperature zones. The crosslinking of neighboring chains with aluminum provides the instability to the system.

### 3.4. Horizontal Burning Test

The HBT of PMMA composites with AlI_3_ was conducted according to ASTM D 4986. One end of the PMMA blend stirp was attached horizontally to a stand, and the other end of the strip was ignited with a flame. The time of flame move from the first reference, marked as A (25 mm from the end) to another reference, marked as B (100 mm from the same end), was noted. The burning rate was determined by the following equation:HBR = Distance of the strip between two reference points/time of burning(1)

The neat PMMA burns within 15 s with the highest burning rate of 6.66 mm/s. The PMMA composite M5 strip of the same length (100 mm) burns in 67 s with the lowest burning rate of 1.49 mm/s. The burning rate of composite M5 containing 10% of AlI_3_ decreased five times as much as compared to neat PMMA (M), whereas the effectiveness was very even for the small proportion of AlI_3_ given in Table 1. This confirmed the retarding effect caused by the AlI_3_ additive to PMMA (M1–M5). The trend is a linear one, i.e., the higher the proportion of AlI_3_ additives, the smaller the rate of burning, and vice versa (Figure 5, Appendix A). HBT confirmed that AlI_3_ produced a flame-retarding effect.

### 3.5. Morphology Analysis

SEM study is helpful to visualize the morphology, diameter of dispersed particles, and composition of the surface. SEM image of neat polymer M is indicated in Figure 6A, that it has a uniform well-smoothed surface and the well-dimensioned diameter of the particle. Its surface has no prominent damage or breakage. SEM analysis of polymer M and composite M5 is indicated in Figure 6B, showing that additive All_3_ spreads uniformly at nanoscale throughout the polymer matrix. The additive nanoparticle is spherical. The SEM images show that the composite surface is a smooth, homogeneously dispersed additive, and no aggregates appeared (Figure 6 and Appendix A).

## 4. Conclusions

A specially designed pyrolytic assembly was used to condense fragments produced by thermal degradation of PMMA with AlI_3_. AlI_3_ changed the degradation mechanism, producing fewer toxic gases confirmed by GC-mass spectrometry. The thermal decomposition of composite M5 is completed in two stages. The first step of degradation of M5 starts at 110 °C and ends at 270 °C a with an 18% weight loss observed. The second-step decomposition starts at 270 °C and ends at 470 °C with a 67% weight loss occurring with 15% residue. It was found that by increasing the percentage of AlI_3_ as an additive in the PMMA composite from 2% to 10% (M1–M5), the initial degradation temperature T_0_ decreased from 250 to 110 °C, and a weight loss with a loss of weight of 10–18% occurred. The burning rate of PMMA composite M5 with 10% of AlI_3_ additive decreased five times as much with residual weight increases from 3% to 15% as compared with pure PMMA.

## Figures and Tables

**Figure 1 materials-14-07036-f001:**
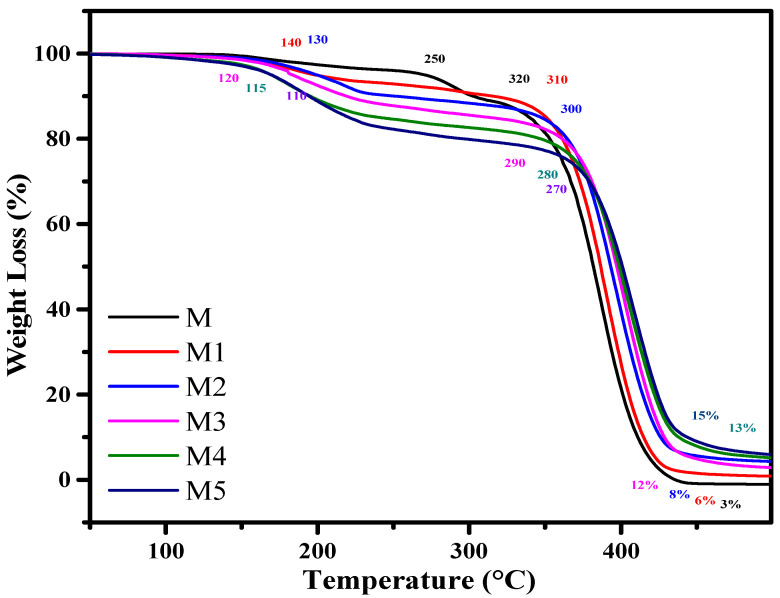
TGA curves for neat PMMA (M) and composites (M1–M5).

**Figure 2 materials-14-07036-f002:**
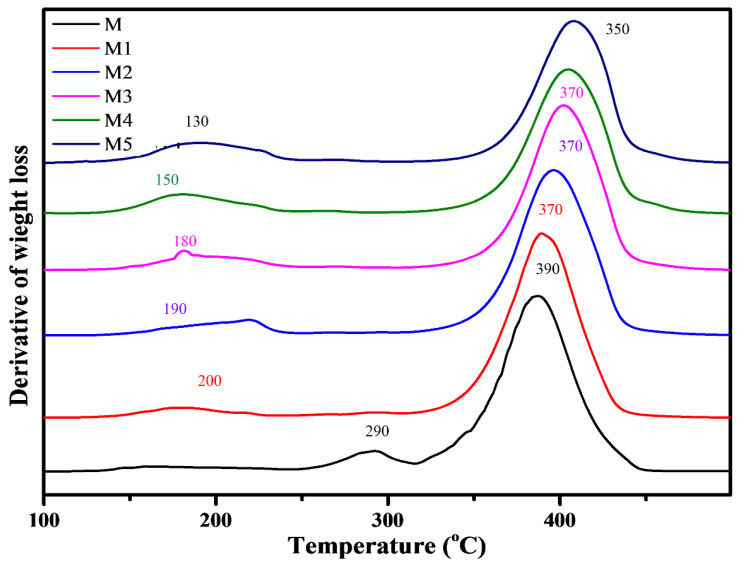
DTG curves for neat polymer M and composites (M1–M5).

**Figure 3 materials-14-07036-f003:**
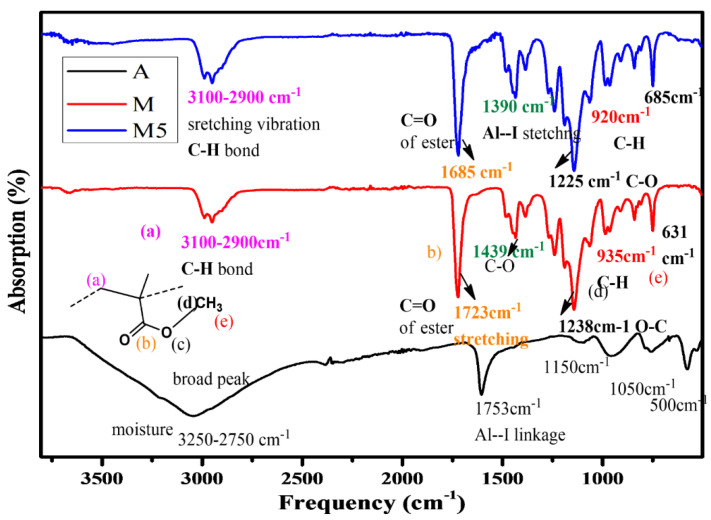
Infrared spectra of M, Additive A, and composite M5.

**Figure 4 materials-14-07036-f004:**
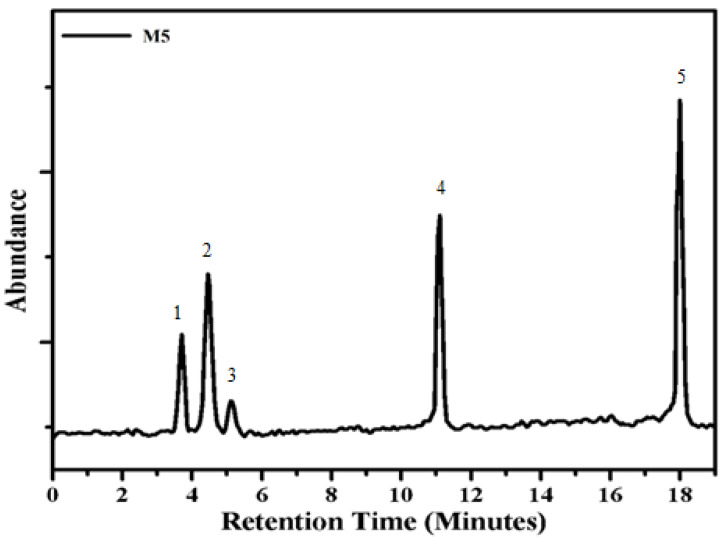
GC-MS chromatogram of composite M5.

**Figure 5 materials-14-07036-f005:**
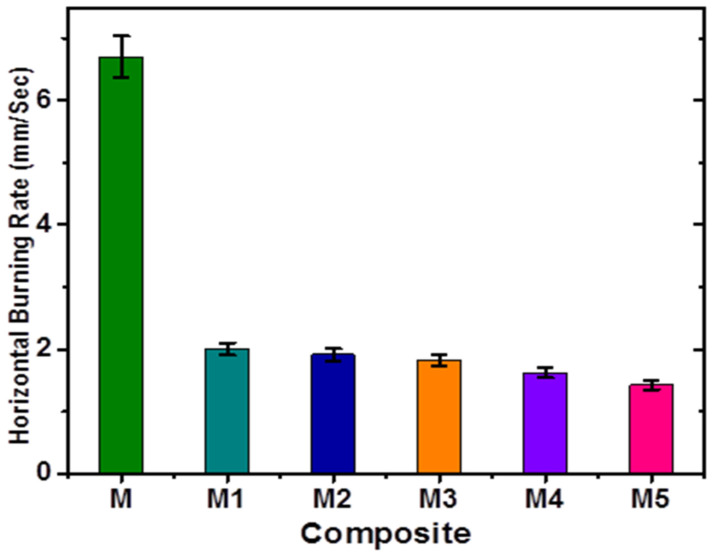
The horizontal burning rate of composites M–M5.

**Figure 6 materials-14-07036-f006:**
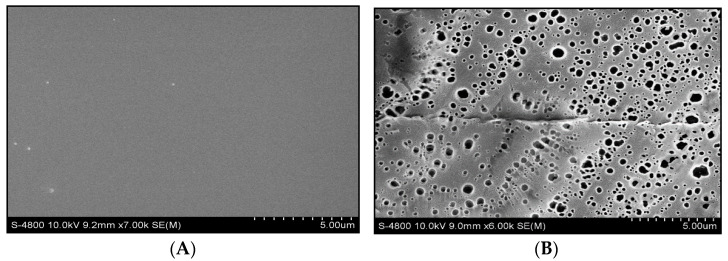
(**A**) SEM image of PMMA. (**B**) SEM image of composite M5.

**Table 1 materials-14-07036-t001:** Formulation of the composite.

Ingredients	Composite Mix Number					
	M	M1	M2	M3	M4	M5
PMMA (%)	100	98	96	94	92	90
AlI_3_ (%)	-	2	4	6	8	10

**Table 2 materials-14-07036-t002:** Comparative study of TGA data for PMMA, AlI_3_, and composite. (M1–M5).

Sample	All_3_%	Stage	Temperature Range (°C)	Weight Loss (%)	T_0_	T_25_	T_50_	T_max_
M	0	I	250–320	10				
		II	320–450	90	250	390	400	450
M1	2	I	140–250	10	140	380	390	460
		II	250–460	87				
		Resi.	>460	3				
M2	4	I	130–240	12	130	370	380	460
		II	240–460	82				
		Resi.	>470	6				
M3	6	I	120–250	14	120	360	370	470
		II	250–470	78				
		Resi.	>470	8				
M4	8	I	115–260	16	115	340	350	470
		II	260–470	72				
		Resi.	>470	12				
M5	10	I	110–270	18	110	320	390	475
		II	270–475	67				
		Resi.	>475	15				
AlI_3_		I	60–160	20	60	150	200	490
		II	165–490	72				
		Resi.	>490	8				

T_0_ = Initial weight loss T, T_25_ = 25% weight loss T, T_50_ = 50% PMMA degradation T, and T_max_ = maximum weight loss T.

**Table 3 materials-14-07036-t003:** GC-MS results of composite M5 after heating at 400 °C.

Peak	R.T (mint.)	MS (m/e)	Fragments Identified
1	3.9	210, 183, 167, 143, 140, 127	C_3_H_4_OIAl
2	4.8	254, 239, 225, 210, 194, 179, 52	C_8_H_15_OI
3	5.2	253, 252, 221, 193, 180, 167, 40	C_7_H_11_O_2_I
4	11.2	347, 314, 298, 267, 235, 209, 194, 179, 162	C_6_H_9_O_7_IAl
5	18.1	480, 449, 322, 308, 294, 277, 206, 249, 221	C_10_H_15_O_4_I_2_Al

## Data Availability

The data presented in this study are openly available in the article.

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
