# Peer review of "The Effect of AlI3 Nanoadditive on the Thermal Behavior of PMMA Subjected to Thermoanalytical Py-GC-MS Technique"

_materials, 2021, doi:10.3390/ma14227036_

Round 1

Reviewer 1 Report

The manuscript “The Effect of AlI3 Nanoadditive on the Thermal Behavior of PMMA Subjected to Thermoanalytical Py-GC-MS Technique ” was well written. After following revision, the paper can be accepted:

  1. Why is this choice of additives? Why is this choice of additives?
  2. What are the properties of a thin film AlI3?
  3. Is only HBT test enough?
  4. The FTIR, TGA and DTA methods are perfectly described.
  5. Explain Fig. 4
  6. Top up the conclusion. Connect with abstract and point out the results obtained

Author Response

Comment 1. Why is this choice of additives? Why is this choice of additives?

Answer: Thank you very much for the valuable comments provided. Organometallic and Inorganic metallic halides have shown very good flame retarding properties. In continuation of our previous investigations, We have checked the thermal degradation pattern of various polymers in the presence of different inorganic salts and organometallic additives. In this study AlI3 salt was used as a flame retarding additive. Additionally, both are acetone soluble and easy to make blend with Polymethylmethacrylate.

Comment 2: What are the properties of a thin film AlI3?

Answer: PMMA is a completely amorphous vinyl polymer but it possesses high strength and excellent dimensional stability due to its rigid polymer chains. It shows exceptional optical clarity, very good weatherability, and impact resistance. AlI3 additive in a very small amount which does not change the mechanical properties of PMMA but only change degradation mechanism due to generation of iodide free radicals during thermal degradation process. There are a lot of publications on the study of properties of PMMA film. So, in order to increase the interest of readers, unnecessary information has been avoided. We only focus on thermal properties of PMMA composites with additive All3

Comment 3: Is only HBT test enough?

Answer: Thank you very much for the valuable comments provided.  Flammability test was conducted to check the affectivity of AlI3. A horizontal burning test was used to check the flame retarding effect of AlI3 additive on PMMA. This test shows that the burning rate of PMMA in the presence of additives is lower than pure PMMA. We agree that more study would be useful to understand details of interaction and enhancement. At this point we do not have the necessary tool-set to study the structure/interaction. Whereas GC-Mass pyrolytic assembly, TGA, DTA data have been provided in the manuscript to support the thermal study of PMMA

Comment 4: The FTIR, TGA and DTA methods are perfectly described.

Answer: We appreciate the enlightening comment, which is of great help to improve the quality of this manuscript. According to the suggestion, FTIR, TGA and DTA methods have been described in detail in the revised manuscript.

Comment 5: Explain Fig. 4

Answer: Fig. 4 has been explained in detail in the revised manuscript.

Comment 6: Top up the conclusion. Connect with abstract and point out the results obtained

Answer: Conclusion part of the manuscript has been improved according to suggestions by adding results of the studies.

Reviewer 2 Report

The manuscript as it stands does not have the expected levels of:

  1. scientific robustness
  2. novelty
  3. the methodology, collation and analyses of the empirical data is far from satisfactory
  4. the conclusions drawn are not valid the data in hand

Furthermore, the English needs major revision- as it stands the manuscript is not very cohesive and is definitely loosely bound together 

Author Response

Comment 1. scientific robustness

Answer: Inorganic metallic halides have shown very good flame retarding properties. So, We have checked the thermal degradation pattern of various polymers in the presence of different inorganic salts. In this study AlI3 salt was used as a flame retarding additive. Which changes the degradation mechanism and reduces the production of toxic fragments produced during thermal degradation of PMMA.

Comment 2. novelty

Answer: Thank you very much for the valuable comments provided. Main novelty of the study is

  1. In this study AlI3 salt was used as a flame retarding additive. Which changes the degradation mechanism and reduces the production of toxic fragments, produced during thermal degradation of PMMA.

  1. We have developed a pyrolytic assembly used to collect degraded fragments of polymers during thermal degradation study.

  1. Study of the degradation of PMMA mixed film with aluminum triiodide while getting insight into the behavior of the polymeric system in the presence of additives.

Comment 3. the methodology, collation and analyses of the empirical data is far from satisfactory

Answer: We appreciate the enlightening comment, which is of great help to improve the quality of this manuscript. According to the suggestion, Methodology and analysis of data has been improved in the revised manuscript.

Comment 4. the conclusions drawn are not valid the data in hand

Answer: Conclusions part have been improved with results of the studies in the revised manuscript.

Comment 5: Furthermore, the English needs major revision- as it stands the manuscript is not very cohesive and is loosely bound together 

 Answer: Based on reviewer’s suggestions, we have revised the manuscript carefully and tried to avoid any grammatical errors. In addition, we have checked all the references and formatted them strictly according to the materials. We have carefully checked the manuscript and corrected the errors accordingly

Reviewer 3 Report

The manuscript describes a study on the effect of the additive AlI3 on the thermal behavior of PMMA.
Here are some considerations in specific points of the article:
page 12, line 53: analysis methods are used to detect degradation behavior and not reduce polymer degradation.

page 2, section 2.1: please, complete the phrase

page 2 and 3, item 2.3: it is necessary to describe the proportions of polymer and additive used.

page 3, section 2.4: what are the other dimensions of the strip?

page 3, section 2.5: what is the ASTM standard number?

page 3, section 2.6: what is the heating rate applied in the thermal analyzer?

page 3, section 2.6: how were de samples for SEM prepared?

page 4, table 1: please specify the meaning of T0, T25, T50 and Tmax

page 7, line 233: just the fact of being transparent does not mean, itself, that the components are compatible

page 8, lines 242 to 246: the phrase must be in the Materials and Methods section

page 9, Figure 5: please, identify the peaks in the graphic;

page 10, section 3.4: the results fo the burning tests must be presented.

page 10, table 3: this table must be in the Materials and Methods Section

In general, the article must be improved in the form of presentation.

The language should be much improved, as some parts are practically incomprehensible

Author Response

Comment 1. page 12, line 53: analysis methods are used to detect degradation behavior and not reduce polymer degradation.

Answer: We are sorry for the mistake. Correction has been made in the revised manuscript. As “Various analysis methods are used to detect the degradation behavior of polymers”

Comment 2. page 2, section 2.1: please, complete the phrase

Answer: Sorry for the carelessness. Correction have been made in the revised manuscript as

Aluminum triiodide (Aldrich Sigma; 99% purity), the monomer, methyl methacrylate (E. Merck), Ammonium persulphate (APS, E. Merck), methanol (E. Merck), and acetone (E. Merck) were used without purification

Comment 3. page 2 and 3, item 2.3: it is necessary to describe the proportions of polymer and additive used.

Answer: We appreciate the enlightening comment, which is of great help to improve the quality of this manuscript. According to the suggestion, the proportion of polymer and additive have been provided in table 1 in the revised manuscript.

Table 1. Formulation of the composite.

Ingredients

Composite mix number

M

M1

M2

M3

M4

M5

PMMA (%)

100

98

96

94

92

90

AlI3 (%)

-

2

4

6

8

10

Comment 4. page 3, section 2.4: what are the other dimensions of the strip?

Answer: dimension of the strip has been provided in the revised manuscript. Dimensions of the strip are 125 mm X 7 mm X 1 mm (Length =125 mm, Wirth = 7 mm, thickness = 1 mm)

Comment 5. page 3, section 2.5: what is the ASTM standard number?

Answer: HBT of PMMA composites with AlI3 was conducted according to ASTM D 4986. Correction has been made in the revised manuscript.

Comment 6. page 3, section 2.6: what is the heating rate applied in the thermal analyzer?

Answer: thermal instability of the polymer sample notified from 25 to 600 °C temperature in a nitrogen environment at heating rate of 5 °C/min

Comment 7. page 3, section 2.6: how were the samples for SEM prepared?

Answer: The PMMA and their composite were put in the common solvent (acetone) overnight to dissolve completely. The solution was placed in the Teflon molds to firm composite film of PMMA composite with AlI3 and dried for 48 h. The dried film was washed with ethanol and used for SEM analysis.

Comment 8. page 4, table 1: please specify the meaning of T0, T25, T50 and Tmax

Answer: T is the temperature at which initial weight loss started, T25  is the temperature at which 25% weight loss of the PMMA occurred, T50  is the temperature at which 50% polymer degraded, and Tmax  is the temperature at which maximum weight loss of PMMA occurred in TGA study.

Corrections have been made in the revised manuscript as “T0 = T at which initial weight loss started, T25 = T at which 25% weight loss occurred, T50 =T at which 50% PMMA degraded, and Tmax  =T at which maximum weight loss occurred

Comment 9: page 7, line 233: just the fact of being transparent does not mean, itself, that the components are compatible

Answer: we agree with the reviewer’s comments and the confusing statement has been corrected in the manuscript as “Both PMMA and AlIwere soluble in acetone and mixed at the molecular level.”

Comment 10: page 8, lines 242 to 246: the phrase must be in the Materials and Methods section

Answer: According to suggestion, the phrase has been added in the materials and method section as

2.6. Pyrolytic study of PMMA composite

The composite M5 (90% PMMA, 10% AlI3), was subjected to pyrolytic study on heating at high temperature in pyrolytic glass assembly. The volatile degradation products were collected in acetone at -196 °C temperature that was maintained under a nitrogen atmosphere. The separation of degradation products occurred by directly injecting a handsome volume of samples into a GC-MS instrument.

Comment 11: page 9, Figure 5: please, identify the peaks in the graphic;

Answer: Peaks in the graph of Figure 5 has been marked in the revised manuscript as

Comment 12: page 10, section 3.4: the results of the burning tests must be presented.

Answer: The results of horizontal burning test have been provided in the supporting information (Table S1)

Table S2: The horizontal burning rate for PMMA nanocomposite.

Composite

Burning time (s)

Length of strip (mm)

Burning rate (mm/s)

M

15

100

6.66

M1

51

100

1.96

M2

53

100

1.88

M3

56

100

1.78

M4

61

100

1.63

M5

67

100

1.49

Comment 13: page 10, table 3: this table must be in the Materials and Methods Section

Answer: We sincerely appreciate the valuable suggestion. The Table 3 has been moved to the material and methods section.

Table 1. Formulation of the composite.

Ingredients

Composite mix number

M

M1

M2

M3

M4

M5

PMMA (%)

100

98

96

94

92

90

AlI3 (%)

-

2

4

6

8

10

Comment 14:  In general, the article must be improved in the form of presentation. The language should be much improved, as some parts are practically incomprehensible

Answer: Language of manuscript has been improved, and manuscript is devoid of spelling and grammatical mistakes. We have carefully checked the manuscript and corrected the errors accordingly

Round 2

Reviewer 1 Report

Accept in present form

Author Response

Thank you very much for the comments and suggestions. I really appreciate the enlightening comments, which are very helpful for improving the quality of our manuscript, as well as the important guiding significance to other research.

Reviewer 2 Report

The authors have endevoured to improve the manuscript in accordance with the observations of the reviewers, and hence as it stands attracts enough merit to be published.

Author Response

(The authors gave the same response as above.)

Reviewer 3 Report

The manuscript shows a noticeable improvement. It is ready to be published after a small revision of the language, especially in the last paragraph of the Introduction.

Author Response

We appreciate the enlightening comment, which is of great help to improve the quality of this manuscript. Based on reviewer’s suggestions, we have revised the manuscript carefully and tried to avoid any grammatical errors. We have studied the comments carefully and made corrections which we hope to meet with approval.